# Effectiveness of Mantra-Based Meditation on Mental Health: A Systematic Review and Meta-Analysis

**DOI:** 10.3390/ijerph19063380

**Published:** 2022-03-13

**Authors:** Yolanda Álvarez-Pérez, Amado Rivero-Santana, Lilisbeth Perestelo-Pérez, Andrea Duarte-Díaz, Vanesa Ramos-García, Ana Toledo-Chávarri, Alezandra Torres-Castaño, Beatriz León-Salas, Diego Infante-Ventura, Nerea González-Hernández, Leticia Rodríguez-Rodríguez, Pedro Serrano-Aguilar

**Affiliations:** 1Canary Islands Health Research Institute Foundation (FIISC), 38109 Tenerife, Spain; amado.riverosantana@sescs.es (A.R.-S.); andrea.duartediaz@sescs.es (A.D.-D.); vanesa.ramosgarcia@sescs.es (V.R.-G.); anatoledochavarri@sescs.es (A.T.-C.); atorrcas@sescs.es (A.T.-C.); beatriz.leonsalas@sescs.es (B.L.-S.); diego.infanteventura@sescs.es (D.I.-V.); leticia.rodriguezrodriguez@sescs.es (L.R.-R.); 2Evaluation Unit (SESCS), Canary Islands Health Service (SCS), 38109 Tenerife, Spain; lilisbeth.peresteloperez@sescs.es (L.P.-P.); pseragu@gobiernodecanarias.org (P.S.-A.); 3Spanish Network of Agencies for Health Technology Assessment for the National Health Service (RedETS), 28071 Madrid, Spain; 4Research Network on Health Services in Chronic Diseases (REDISSEC), 38109 Tenerife, Spain; ngonzalez@kronikgune.org; 5Osakidetza Basque Health Service, Barrualde-Galdakao Integrated Health Organisation, 48960 Galdakao, Spain; 6Kronikgune Institute for Health Services Research, Ronda de Azkue 1 torre del Bilbao Exhibition Centre, 48902 Barakaldo, Spain

**Keywords:** mantra-based meditation, mental health, systematic review, meta-analysis

## Abstract

Background: Meditation is defined as a form of cognitive training that aims to improve attentional and emotional self-regulation. This systematic review aims to evaluate the available scientific evidence on the effectiveness and safety of mantra-based meditation techniques (MBM), in comparison to passive or active controls, or other active treatment, for the management of mental health symptoms. Methods: MEDLINE, EMBASE, Cochrane Library, and PsycINFO databases were consulted up to April 2021. Randomised controlled trials regarding meditation techniques mainly based on the repetition of mantras, such as transcendental meditation or others, were included. Results: MBM, compared to control conditions, was found to produce significant small-to-moderate effect sizes in the reduction of anxiety (g = −0.46, IC95%: −0.60, −0.32; I^2^ = 33%), depression (g = −0.33, 95% CI: −0.48, −0.19; I^2^ = 12%), stress (g = −0.45, 95% CI: −0.65, −0.24; I^2^ = 46%), post-traumatic stress (g = −0.59, 95% CI: −0.79, −0.38; I^2^ = 0%), and mental health-related quality of life (g = 0.32, 95% CI: 0.15, 0.49; I^2^ = 0%). Conclusions: MBM appears to produce small-to-moderate significant reductions in mental health; however, this evidence is weakened by the risk of study bias and the paucity of studies with psychiatric samples and long-term follow-up.

## 1. Introduction

Mental health is a key component in people’s quality of life and represents a central aspect of national and international action plans on health policies, hence the importance of evaluating different interventions aimed at improving the prevention and treatment of mental symptoms and disorders, including meditation techniques [1]. Meditation can be defined as a cognitive training aimed to improve attentional and emotional self-regulation [2]. This is a general term that includes a wide and heterogeneous set of attention and concentration practices, mostly derived from ancient traditions [3]. In general, these techniques aim to increasing awareness and exercises voluntary control over mental processes in order to achieve better understanding of one’s own mental activity and develop optimal states of psychological well-being [3,4]. Currently, there are a wide variety of meditation practices which accentuate the problem of listing, differentiating and theoretically defining them [5,6].

Mantra-based meditation (MBM) generally involves the continuous repetition of a word, phrase, or set of syllables (either silently or aloud) with or without religious/spiritual content. The sound of a mantra within meditation has been suggested to act as an effective vehicle to override mental speech, which is the predominant form of conscious for most people, and to redirect those negative or intrusive automatic thought that perpetuate distress [7]. While MBM has generally been categorised as a type of focused attention meditation, it has been suggested that the repetition of a sound, word, or sentence makes MBM unique given its specific voluntary linguistic production, rather than the natural production of body-related sensations or the focus on external physical objects [8]. Transcendental meditation (TM) is one of the most common forms of MBM. It consists on the repetition of non-religious mantras and seeks to settle the mind in calmer levels of thought until reaching a silent state of transcendental consciousness [9]. Within this MBM group are also included other mantra-based meditations (OMBM) such as Benson relaxation, Jyoti meditation, clinically standardised meditation (CSM), and ACEM meditation, among others.

In the last years, there has been a growing interest in the study of the effectiveness and safety of MBM. Indeed, systematic reviews (SR) and meta-analyses (MA) have been conducted for both physical and mental conditions [7,10,11]. These studies have shown that MBM is associated with reductions in systolic and diastolic blood pressure [11], is more effective than usual treatment on reducing trait-anxiety [10], and can improve mental health and negative affectivity in non-clinical populations [7]. Several randomised-controlled trials (RCT) have been conducted since the publication of these reviews [12,13,14,15]. Moreover, potential adverse events during or after meditation practices have received very limited attention [16], and therefore it is necessary to assess whether MBM is a safe technique to be applied in the mental health field. Accordingly, this SR aims to identify, critically evaluate, and synthesise all the available scientific evidence on the effectiveness and safety of MBM techniques for the management of mental health symptoms.

## 2. Materials and Methods

The review protocol has been prospectively registered in PROSPERO (CRD42021243696). The Preferred Reporting Items for Systematic Reviews and Meta-Analyses (PRISMA) statement guided the current research (see Appendix A) [17].

### 2.1. Literature Search and Study Selection

The following databases were searched from inception date to April 2021: MEDLINE, EMBASE, COCHRANE, and PsycINFO. The search strategy was not specific for MBM, but included more terms related to other meditation techniques in addition to MBM. It included “meditation” as topic, as well as specific terms (for instance: (Focused attention or Mantra or Yantra or Breath or Transcendental or Zazen or Tonglen or Dzogchen or Metta) adj2 (meditation* or therap* or treatment* or intervention* or training*)).ti,ab.) (see Appendix A). The terms were combined and adapted to each search engine using sensitive filters for RCTs. 

The study selection was carried out independently by two reviewers, and disagreements were resolved by discussion with the participation of a third reviewer. First, the titles and abstracts were evaluated to determine their relevance to the selection criteria. Subsequently, the full texts of the relevant articles were reviewed and selected on the basis of the full inclusion criteria. References included were scanned for further RCTs.

### 2.2. Inclusion Criteria

The inclusion criteria were: (a) RCTs; (b) studies published in peer reviewed journals; (c) studies including general population, healthy participants, or people with physical or mental diseases/symptoms/complaints; (d) measuring outcomes related to safety (i.e., adverse effects, withdrawals due to adverse) and/or effectiveness (i.e., reduction of mental health symptoms, including anxiety, depression, distress, post-traumatic stress disorder (PTSD), burnout, insomnia, psychotic symptoms, behavioural disorders, suicidality); (e) targeting an intervention based on meditation techniques mainly based on the repetition of mantras; (f) including comparators such as waiting list, no intervention/usual care, active control (e.g., health education, discussion groups, physical exercise), and other evidence-based treatments (e.g., pharmacological treatment, psychotherapy, or relaxation techniques); and (g) articles written in English or Spanish without any date, gender, age of the participants, or social context restrictions.

We excluded mindfulness meditation, techniques exclusively based on breathing or scanning of corporal sensations, loving-kindness meditation, self-compassion-based meditation, religious/spiritual meditation (if mantra is not the core component of the program), and body–mind techniques (e.g., yoga, tai chi).

### 2.3. Data Extraction and Assessment of Methodological Quality

Studies characteristics and risk-of-bias were extracted and assessed, independently, by two reviewers. A third review author was consulted regarding any discrepancies, and these were resolved by discussion until consensus was reached. Data extraction was carried out in an Excel^®^ file reflecting the information of each study (e.g., dates of data collection), sample characteristics (e.g., age, gender, diagnosis), study design (e.g., comparison condition), intervention details (e.g., number sessions, format), and intervention components. Studies were assessed for methodological quality according to the Cochrane Collaboration’s tool for assessing risk of bias, the Cochrane risk-of-bias tool (RoB 2) [18]. The high risk of bias rating in any of the assessed domains or the uncertain risk of bias rating in three or more domains leads to qualification of the study as having an overall high risk of bias. None of the included studies were assessed with a low risk of bias. Several included studies evaluated both self-reported (e.g., questionnaires of anxiety, depression, quality of life) and hetero-reported measures (e.g., symptoms observed by the clinician); thus, in these cases, a double evaluation of the risk of bias of the study was carried out, considering each type of outcome measure.

### 2.4. Analysis

An MA using R-studio software (packages meta [19], metafor [20], and dmetar [21]) was performed for each outcome measure when data were available. Since most studies assessed outcomes between 1 and 6 months after starting the meditation program, when a study assessed them at several time points (either during the intervention or follow up), we selected the one closest to 4 months. The between-group standardised difference (Hedges g) was extracted from each study. If it was not available or could not be calculated from the reported data, the difference in the change from baseline to follow-up was extracted. In both cases, estimates calculated by intention-to-treat and/or adjusted for confounders were included if available. Several studies did not report standard deviations (SD); these were imputed (separately for the intervention and control groups), calculating the average of the mean/SD ratio obtained in the remaining studies (or only in those that applied the same type of meditation when there were five or more). Sensitivity analyses were carried out excluding these studies.

The MA was performed using the inverse variance method. A random effects model was applied using the Sidik–Jonkman method as a tau estimator [22]. Statistical heterogeneity between the different studies included in the MA were assessed using the Higgins I^2^ value [23]. For each MA, two-tailed 95% prediction intervals were calculated. The following subgroup analyses were carried out: type of meditation (TM vs. OMBM vs. mixed), type of scores (pre-post change vs. post-scores), type of control group (waiting list, no intervention/usual care, active control), type of participants (general population, students, clinical samples), length of follow up (<1 month vs. 1–4 months vs. >4 months; in this analysis, the last follow up measurement was used in studies with more than one assessment) and year of publication (before/after the median year).

Publication bias was visually analysed using the funnel plot and statistically with the Eggers’ test [24], when there were eight or more studies. The results of the studies that could not be included in MA are described narratively.

## 3. Results

### 3.1. Results of the Search

The initial search in the electronic databases yielded 5982 references. After removing duplicates and screening by title and abstract, 278 full-text articles were assessed for eligibility. Three additional records were identified through manual searches and citation lists. Fifty-one studies, reported in 52 references, were finally included (see Figure 1). So et al. (2001) [25] included three studies with different samples, and thus it was analysed as three independent studies.

### 3.2. Characteristics of Included Studies

The characteristics of included studies are provided in Appendix A. The types of meditation used were TM (27 studies), OMBM (24 studies in 25 references), and mixed meditations (2 studies). Twenty-two studies recruited people from the general population, 12 included students and 31 included clinical samples.

### 3.3. Methodological Quality 

None of the 52 studies was assessed at low risk of bias (see Appendix A). Most were rated as high risk of bias and seven were rated as uncertain bias. Several studies did not provide information on the blinding procedures carried out, but given the characteristics of the interventions, the meditation instructors could not be blinded to the conditions of the groups; therefore, no study had a low risk of bias in domain 2.

### 3.4. Effects of the Intervention

#### 3.4.1. Anxiety

***Meditation versus control group***. Twenty-five studies (reported in 23 articles, *n* = 1825) assessed self-reported anxiety: 12 studies (in 10 references) evaluated TM [25,26,27,28,29,30,31,32,33,34] and 13 OMBM [35,36,37,38,39,40,41,42,43,44,45,46,47]. 

Participants included students (10 studies), general population (7 studies), and clinical (8 studies) population (one study with male prison inmates without clinical diagnosis [31] was included in the latter subgroup since it assessed post-traumatic stress as the main measure). SD were imputed in five studies [26,34,37,45,47], and three only reported change data. The exclusion of these studies, individually or jointly, did not substantially modify the results. 

The overall result was significantly to the intervention (g = −0.46, IC95%: −0.60, −0.32; I^2^ = 33%; prediction interval: −0.99, 0.07). The funnel plot showed a symmetric distribution of the studies, with no evidence of publication bias (Eggers’ test *p* = 0.68) (see Appendix A). Subgroup analyses did not show significant differences by type of meditation (see Figure 2), type of control, population, publication year, or length of follow-up (see Appendix A). 

Another four studies (in five references) with OMBM [12,48,49,50,51] did not report numerical data that could be included in the MA. Vasudev et al. (2016) [50,51], in elderly people with depression (*n* = 51), obtained a significant effect of the intervention compared to the usual treatment at 3 months (F = 2.95, *p* < 0.05). Dunne et al. (2019) [12] (*n* = 58) performed a study with emergency personnel and people with back pain, reporting a significant reduction in the meditation group, but they did not perform an inter-group contrast. The remaining two studies did not obtain significant results. 

***Meditation versus relaxation therapy***. Nine studies included a relaxation group as comparator [26,34,43,44,45,47,52,53,54]. Only one of them [34] reported significant differences, in favour of meditation (*p* < 0.02 for interaction). A MA was performed with four studies [43,44,52,54] (*n* = 163), since the remaining did not report any SD, and imputations were ruled out due to the low number of studies and participants. The result was not significant (g = −0.18, 95% CI: −0.51, 0.15; I^2^ = 0%; prediction interval: −1.01, 0.65) (see Figure 3).

***Meditation versus psychological therapy*****.** Two studies compared meditation with psychotherapy. Brooks et al. (1986) [55], including veterans of the Vietnam War, obtained a strong effect in favour of TM, compared to different types of psychotherapy together (g = −1.75, 95% CI: −2.88, −0.62). Jong et al. (2019) [52], in children with headaches, did not obtain significant differences.

#### 3.4.2. Depression

***Meditation versus control group*****.** Twenty-two studies (reported in 23 articles, *n* = 1616) with data on self-reported depression were included in the MA: 13 evaluated TM [13,14,26,27,29,30,31,32,33,56,57,58,59]; 8 OMBM [36,38,40,42,45,46,50,51,60]; and 1 [61] evaluated a combination of mantra-based meditation, Tibetan sounds, and attention focused on breathing. Thirteen studies included clinical samples, and nine included adults from the general population. Nine studies only provided data on change in scores. 

A significant effect of meditation was found (g = −0.33, 95% CI: −0.48, −0.19; I^2^ = 12%; prediction interval: −0.82, 0.16). After excluding the study that evaluated a mixed meditation [61], the result did not substantially change (g = −0.34, 95% CI: −0.49, −0.19; I^2^ = 14%; prediction interval: −0.84, 0.16) and the subgroup difference was not significant. The effect was stronger for TM (see Figure 4). The remaining subgroup analyses were not significant (see Appendix A). Regarding length of follow up, studies with more than 4 months of follow-up obtained a small effect (g = −0.28, 95% CI: −0.54, 0.00; I^2^ = 0%). The funnel plot does not show evidence of publication bias (Eggers’ test *p* = 0.98) (see Appendix A).

One study (*n* = 178) could not be included in the MA because it reported medians [49]. A significant difference was observed at 2 months, in favour of meditation compared to the delayed intervention group.

***Meditation versus relaxation therapy*****.** Four studies compared TM [26,52] and OMBM [45,54] versus progressive muscle relaxation, but available data discarded an MA. Lehrer et al. (1983) [45] observed a better result for the relaxation group at 5 weeks (*p* < 0.04), but not at 9 months. The other three studies did not obtain significant differences, with follow up ranging from 1 to 9 months.

***Meditation versus psychotherapy*****.** Five studies compared meditation with psychological treatments [14,15,26,52,55]. Significant differences were only observed in Brooks et al. (1986) [55], favouring TM compared to different types of psychotherapy together. Two studies [26,52] did not report numerical data that could be included in the quantitative synthesis. A MA of the three studies (*n* = 327) with usable data was not significant (g = −0.67, 95% CI: −1.70, 0.37; I^2^ = 75%; prediction interval: −13.36, 12.03) (see Figure 5).

#### 3.4.3. Stress

***Meditation versus control group*****.** Fourteen studies (*n* = 1078) were included: six TM [29,30,31,56,57,59] and eight OMBM [35,36,38,42,46,60,62,63]. We excluded measures that are named in some articles as psychological distress, but that assess general psychopathology (e.g., General Health Questionnaire (GHQ), Brief Symptoms Inventory-18 (BSI-18); see Section 3.4.8). Nine studies included adults from the general population and five included clinical samples. For three studies [29,30,56], the difference in change in scores was entered into the MA; its exclusion did not substantially modify the results. 

The MA yielded a result significantly favourable to the intervention (g = −0.45, 95% CI: −0.65, −0.24; I^2^ = 46%; prediction interval: −1.11, 0.22) (Figure 6). The subgroup analysis by type of control was not significant, although only the comparisons with the waiting list group (g = −0.64, 95% CI: −0.94, −0.34; I^2^ = 44%) and no intervention/usual care (g = −0.57, 95% CI: −1.13, −0.01; I^2^ = 65%) obtained a significant result (see Appendix A). Analyses by population types and follow-up were not significant. The effect was only significant in healthy adults (g = −0.55, 95% CI: −0.82, −0.29; I^2^ = 55%) but no in clinical samples (g = −0.24, 95% CI: −0.49, 0.01; I^2^ = 0%) (see Appendix A). Studies with follow up longer than 4 months yielded a lower, non-significant result (g = −0.25, 95% CI: −0.56, 0.06; I^2^ = 21%) (see Appendix A). The funnel plot did not show evidence of publication bias (Eggers’ test *p* = 0.73) (see Appendix A).

Two studies could not be included in the MA. Severtsen et al. (1986) [64] included only five participants per group, showing a strong difference in stress baseline scores between meditation group and aerobic exercise group. Manocha et al. (2011) [49] reported a significant effect of TM at 2 months (median change −37.0 vs. −17.5, *p* = 0.026).

#### 3.4.4. Post-Traumatic Stress Disorders’ Symptoms

***Meditation versus control group*****.** Six studies (*n* = 513) that used the PTSD checklist (PCL) were included in the MA. Four studies applied TM (two with prison inmates [31,65] and two with veterans [14,27]), whereas two studies applied OMBM in veterans [15,40]. The result was significantly favourable to the intervention (g = −0.59, 95% CI: −0.79, −0.38; I^2^ = 0%; prediction interval: −1.00, −0.17) (see Figure 7). Despite showing a very intense effect (g = −8.75, 95% CI: −10.81, −6.70), the study by Rees et al. (2013) [66] was not included in the MA because it showed a high risk of selection bias (59% of the participants assigned to the meditation group did not start the program, and a paired sample design was subsequently carried out). 

Four studies conducted a clinical assessment using the Clinically Administered PTSD Scale (CAPS) [14,27,40,67]. The overall result was significantly favourable to the intervention (g = −0.45, 95% CI: −0.72, −0.18; I^2^ = 0%; prediction interval: −1.33, 0.42) (see Figure 8). 

***Meditation versus psychotherapy*****.** Four studies (*n* = 464) (one of them with two subsamples) that used the PCL were included in the MA. Three studies applied TM [14,55,68] and two studies applied OMBM in veterans [15,68]. The result was significantly favourable to the intervention (g = −0.40, 95% CI: −0.79, 0.00; I^2^ = 27%; prediction interval: −1.69, 0.89) (see Figure 9).

#### 3.4.5. Burnout

***Meditation vs. control group*****.** Four studies with small samples evaluated the effect of TM [56] and OMBS [12,35,62] with the Maslach Burnout Inventory (MBI) on work stress in teachers and health emergency personnel. MA was discarded due to data heterogeneity or unavailability. Elder et al. (2014) [56] (*n* = 40) obtained a significantly favourable effect for the intervention at 3 months (Glass Delta = 0.40, *p* = 0.018). Anderson et al. (1999) [35] (*n* = 91) obtained a significant effect at 9 months in the emotional exhaustion subscale (Glass’s Delta = 1.78, *p* < 0.001). Oman et al. (2006) [62] (*n* = 61) did not obtain significant effects at 19 weeks in any of the MBI subscales (*p*-values > 0.16). Finally, Dunne et al. (2019) [12] (*n* = 51) observed a significant reduction of the emotional exhaustion subscale on the meditation group, but they did not performed the inter-group contrast.

#### 3.4.6. Mental Health-Related Quality of Life

***Meditation vs. control group*****.** A total of nine studies (*n* = 592) were included in the MA, four with TM [29,57,59,69], four with OMBM [40,42,46,62], and one with a mixed type of meditation [61]. Studies that used the Quality-of-Life Enjoyment and Satisfaction Questionnaire (Q-LES-Q) were not included, since this instrument includes non-health-related quality of life dimensions. Two studies included healthy adults [29,62] and seven included clinical samples. The global result was significantly favourable to meditation (g = 0.32, 95% CI: 0.15, 0.49; I^2^ = 0%; prediction interval: 0.07, 0.57) (see Figure 10). Subgroup analyses were not significant (see Appendix A). The funnel plot shows a symmetric distribution of the studies, with no evidence of publication bias (Eggers’ test *p* = 0.61) (see Appendix A).

***Meditation vs. psychotherapy.*** One study (*n* = 173) [15] did not find significant differences at 2 months between OMBM and Present-Centred Therapy in veterans with PTSD.

#### 3.4.7. Sleep Quality

***Meditation vs. control group*****.** Four studies evaluated sleep problems. Two studies obtained significant results favouring TM versus control groups, one with incarcerated men (g = −0.67, 95% CI: −1.01, −0.34) [31] and one with people with PTSD (g = −0.78, 95% CI: −1.43, −0.14) [27]. One study that applied a mixed meditation in women with breast cancer did not observe significant differences after 10 weeks (*p* = 0.29) [61]. Finally, another study [15] in people with PTSD obtained a significant benefit of OMBM at 2 months compared with psychotherapy (g = −0.36, 95% CI: −0.66, −0.06).

***Meditation vs. psychotherapy*****.** One study (*n* = 173) [15] obtained a better result at 2 months for OMBM compared to Present-Centred Therapy in veterans with PTSD (g = −0.36, 95% CI: −0.66, −0.06).

#### 3.4.8. General Psychopathology

***Meditation vs. control group*****.** Three studies (*n* = 188) with clinical samples [13,63,67] were included in the MA of general psychopathology, yielding a significant benefit of meditation (g = −0.57, 95% CI: −0.87, −0.27, I^2^ = 0%; prediction interval: −2.56, 1.42) (see Figure 11). Lehrer et al. (1983) [45], not included in the MA, using the SCL-90 questionnaire, obtained a significant reduction in the OMBM group (*p* < 0.01) but not in the control group.

#### 3.4.9. Substance Consumption

***Meditation vs. control group*****.** Three studies evaluated the effectiveness of meditation on the consumption of alcohol [70,71] and tobacco [72]. In the study by Tuab et al. (1994) [71] (66 men with alcoholism), the TM group showed a significantly lower percentage of days without drinking than the usual treatment at 6, 12, and 18 months (76.3% vs. 45.7% in the latter period, *p* < 0.05). Murphy et al. (1986) [70], in 60 university students with high social consumption of alcohol, did not find a significant effect of clinically standardised meditation (a type of OMBM) at the end of treatment or after 6 weeks of follow-up. Ottens (1975) [72] (*n* = 36) evaluated the effectiveness of TM compared to a discussion group and no treatment with university students. The first two groups reduced their consumption significantly compared to the control at 10 weeks (F = 4.84, *p* < 0.01), without significant differences between them.

### 3.5. Safety

Only four studies explicitly reported information on the safety of meditation, two on TM [57,73] and two on OMBM [46,50]. None of them observed adverse events that could be related to the application of meditation techniques.

## 4. Discussion

This SR has identified a considerable number of studies that assessed the effectiveness of MBM on mental health outcomes. However, apart from the studies focused on PTSD, only a few of them included participants with other confirmed or probable mental disorders. Therefore, results are mainly in reference to people from the general population (e.g., students, adults, employees) or with somatic diseases. Approximately half of the studies applied TM, and anxiety was the most studied variable. 

The comparative results show that MBM produces significant small-to-moderate effect sizes in the reduction of anxiety, depression, stress (including post-traumatic), and mental health-related quality of life, with low or moderate statistical heterogeneity across outcomes. MBM has also shown significant benefits on burnout, sleep problems, general psychotherapy, and substance use, although the number of studies assessing these variables is lower. Previous MA have been limited in number and scope. Orme-Johnson et al. (2014) [10] assessed the effectiveness of TM on anxiety, including 16 studies that yielded a pooled effect of d = 0.62 (95%CI: −0.82, −0.43). Gathright et al. (2019) [11] did not find a significant pooled effect of TM on depression in five studies including patients with cardiovascular disease. However, Lynch et al. (2018) [7] concluded that there is some evidence suggesting that mantra meditation can improve mental health and negative affectivity in non-clinical samples.

These results must be interpreted cautiously due to several factors. First, most included studies present a high risk of bias, and therefore a potential overestimation of effects cannot be ruled out. In addition, although subgroup analyses were mostly non-significant (possibly due to a lack of statistical power in several cases), a consistent pattern was observed across outcomes depending on the control group: effects were stronger in comparison with waiting list or usual care/no intervention, and weaker in comparisons with active controls, suggesting the influence of non-specific factors of the intervention not related to meditation. The SR of Goyal et al. (2014) [74] included only comparators that mirrored in time and attention the meditation program applied, and they did not find significant benefits of MBM on anxiety, depression, or stress, but the number of studies included was lower than in our analyses. Our results on anxiety and depression were significant compared to active controls, although, as commented, they were somewhat less intense (g = −0.39 for anxiety and −0.28 for depression). For stress and quality of life, with four and two studies, respectively, in the active control subgroup, results were non-significant. 

Another relevant effect moderator is the population included. For anxiety and stress, but not for depression and quality of life, effects were lower in clinical samples than in participants from the general population. Furthermore, as commented above, few studies included patients with clinically diagnosed mental disorders. Therefore, apart from PTSD, the effectiveness of MBM in the treatment of mental health disorders remains unclear.

Finally, regarding subgroup analyses, other trends were observed. TM seems to produce greater effects than OMBM. Some studies support the need to differentiate TM from OMBM, since despite using mantras, TM allows the subject to reach a higher state of self-consciousness in which the mantra progressively becomes a secondary experience until its disappearance [1]. On the other side, studies with shorter follow up (<4 months) obtained better results. Unfortunately, studies that applied follow up did not assessed compliance with meditation practice once the “formal” intervention finished, and therefore it is uncertain as to whether this result is due to a decrease in meditation practice over time or a genuine loss of effectiveness. More studies are required to address the underlying mechanisms that elucidate how meditation practice leads to outcome changes in daily life [75].

The available evidence does not show significant differences between MBM and relaxation therapy or psychotherapy, although the number of studies is limited. Orme-Johnson et al. (2014) [10] found that TM was superior to alternative treatments (d = −0.50, 95%CI: −0.70, −0.30), but this analysis combined 10 studies with heterogeneous interventions (e.g., relaxation, psychotherapy, biofeedback, stress management, another type of meditation). 

Regarding safety, only four studies evaluated the adverse effects derived from MBM. This fact possibly reflects the implicit assumption of the perceived safety of these techniques. However, some recent studies and SR suggest that in some meditation practices, such as mindfulness or focused attention, unwanted experiences such as anxiety or depersonalisation may appear during the meditation sessions, even in people without history of mental health problems [16,76,77,78]. These results are relevant and contribute to a balanced perspective of meditation as a practice that may lead to both positive and negative outcomes [16]. Therefore, although serious adverse effects are not likely to occur or remain after discontinuation of practice, future research should systematically evaluate the occurrence of these events.

### Limitations

The main limitation of this SR is the potential non-identification of eligible studies, since we did not search grey literature and the language was restricted to English or Spanish. Another important limitation has to do with the high or uncertain risk of bias of the included studies. Third, since there was a wide between-study variability in the duration of the intervention and follow-up (if applied), we did not evaluate the immediate post-intervention effect, but tried to maximise the homogeneity of the time of measurement selecting the results closest to 4 months. Although the number of included studies was substantial, subgroup analyses were probably underpowered in many cases. As commented above, very few studies included samples with psychiatric disorders. 

## 5. Conclusions

MBM seems to produce significant small-to-moderate reductions (g > 0.40) of the levels of anxiety, stress (including post-traumatic), and general psychopathology, as well as small but also significant improvements in depression and mental health-related quality of life. Other outcomes (i.e., burnout, insomnia, substance consumption) also have shown improvements, but the number of studies assessing them is quite lower. The same occurs with the comparison of MBM with psychotherapy, which has shown no significant differences. This evidence is weakened by the risk of bias of the studies, as well as the scarcity of studies with psychiatric samples and long-term follow up. Regarding safety, potential adverse effects should be systematically investigated.

Meditation techniques aim to improve attentional and emotional self-regulation by means of cognitive training and other practices (e.g., promotion of love, kindness, and compassion; sensorial stimuli; body movement), thus achieving a state of relaxation in the person [2]. They could be a promising resource to improve the mental health of clinical and non-clinical populations, as well as to contribute to reduce the burden of mental disorders and their associated costs for health systems. In the case of MBM, it is easy to learn and practice, and does not require theoretical knowledge or the sharing of specific religious or spiritual beliefs.

## Figures and Tables

**Figure 1 ijerph-19-03380-f001:**
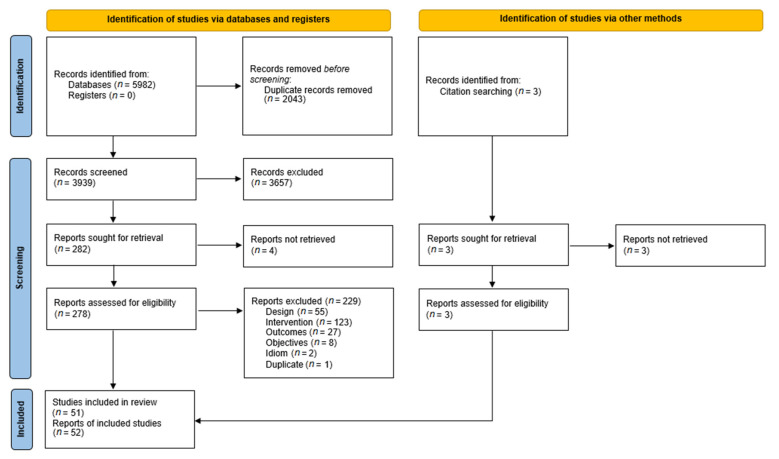
PRISMA flow diagram of the study selection process.

**Figure 2 ijerph-19-03380-f002:**
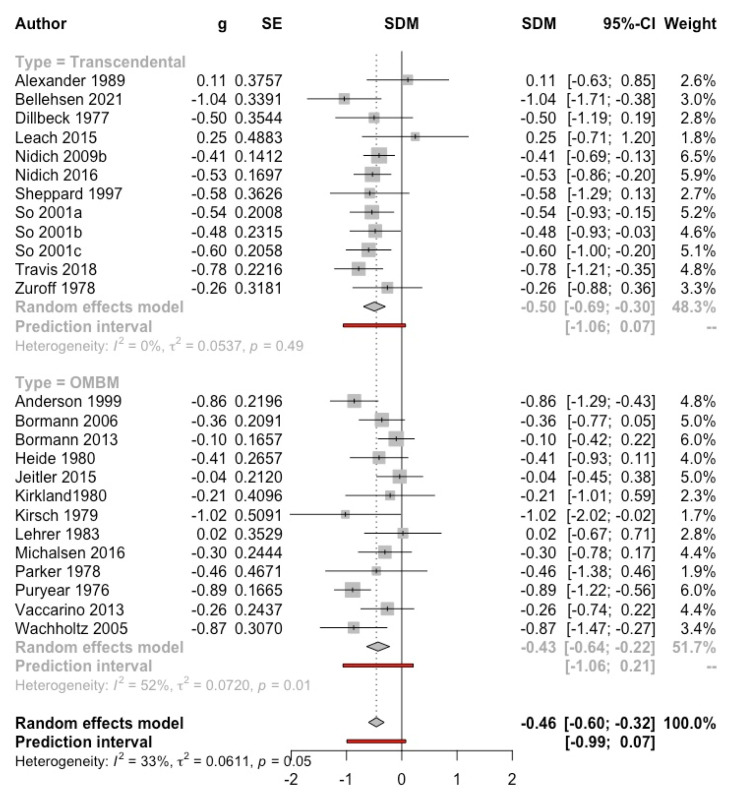
Effect on anxiety (meditation vs. control). Predictive intervals are represented in red.

**Figure 3 ijerph-19-03380-f003:**
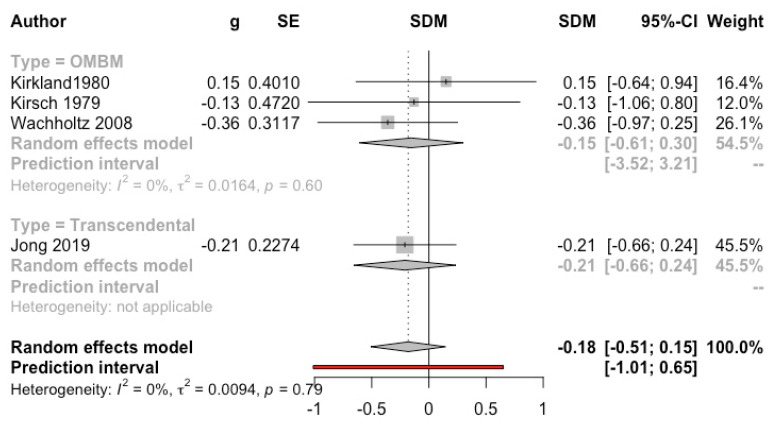
Effect on anxiety (meditation vs. relaxation). Predictive intervals are represented in red.

**Figure 4 ijerph-19-03380-f004:**
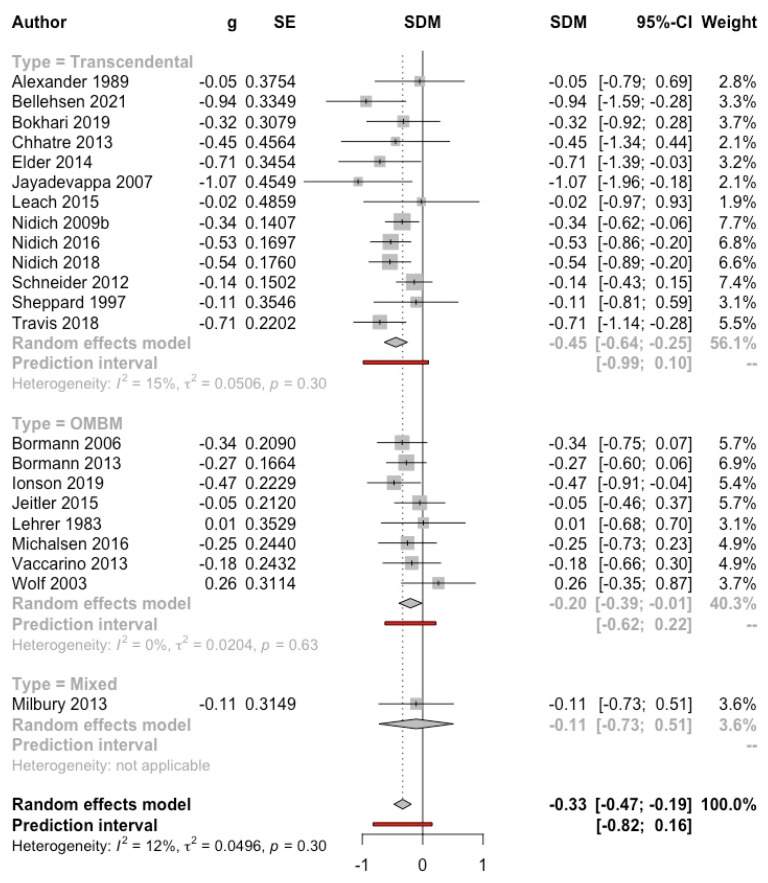
Effect on depression (meditation vs. control). Predictive intervals are represented in red.

**Figure 5 ijerph-19-03380-f005:**
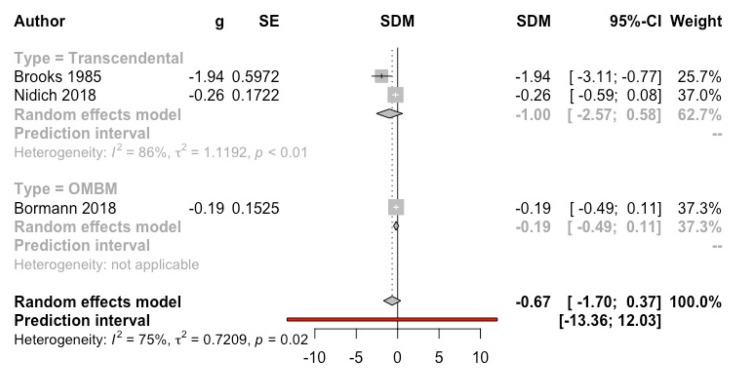
Effect on depression (meditation vs. psychotherapy). Predictive intervals are represented in red.

**Figure 6 ijerph-19-03380-f006:**
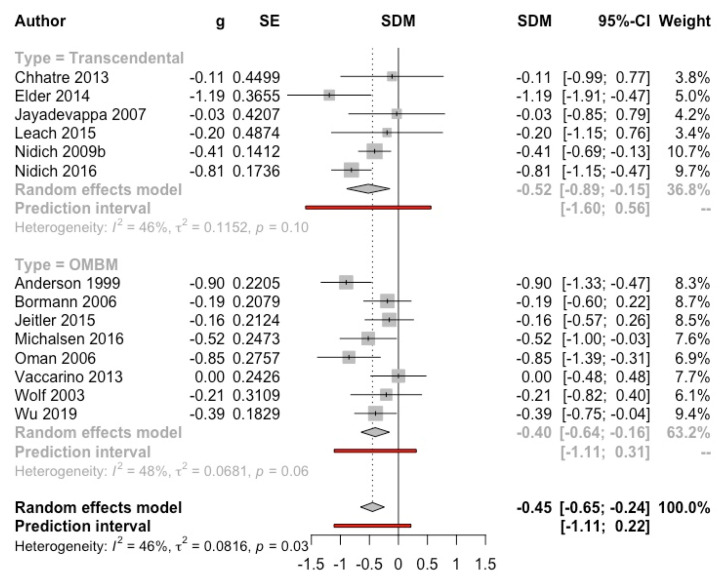
Effect on stress (meditation vs. control). Predictive intervals are represented in red.

**Figure 7 ijerph-19-03380-f007:**
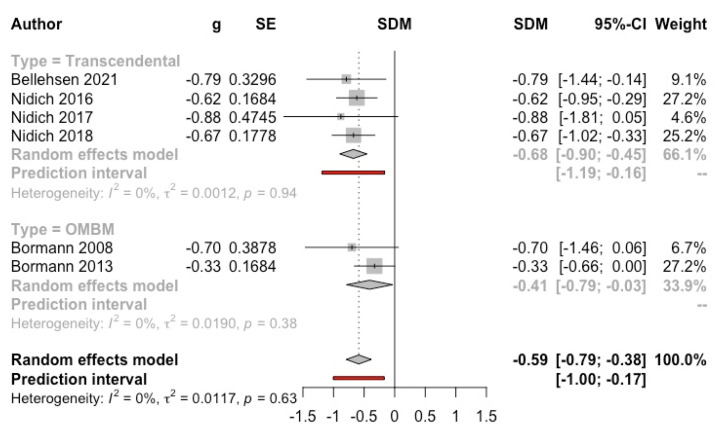
Effect on post-traumatic stress disorder symptoms (meditation vs. control). Predictive intervals are represented in red.

**Figure 8 ijerph-19-03380-f008:**
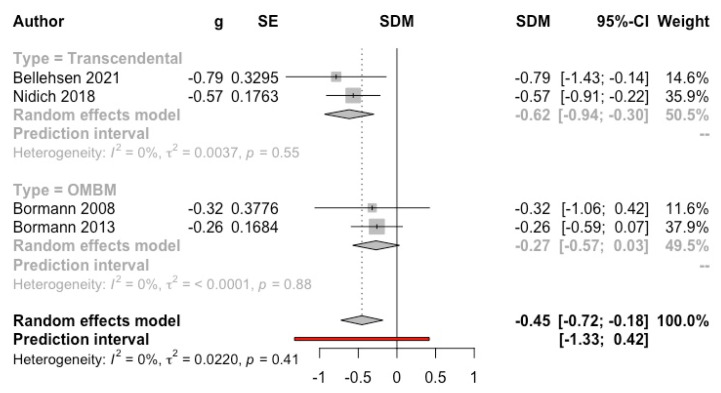
Effect on post-traumatic stress disorder symptoms (clinician-reported) (meditation vs. control). Predictive intervals are represented in red.

**Figure 9 ijerph-19-03380-f009:**
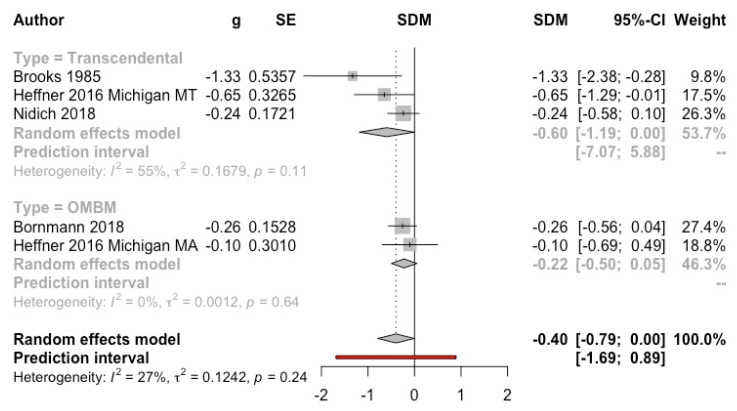
Effect on post-traumatic stress disorder symptoms (meditation vs. psychotherapy). Predictive intervals are represented in red.

**Figure 10 ijerph-19-03380-f010:**
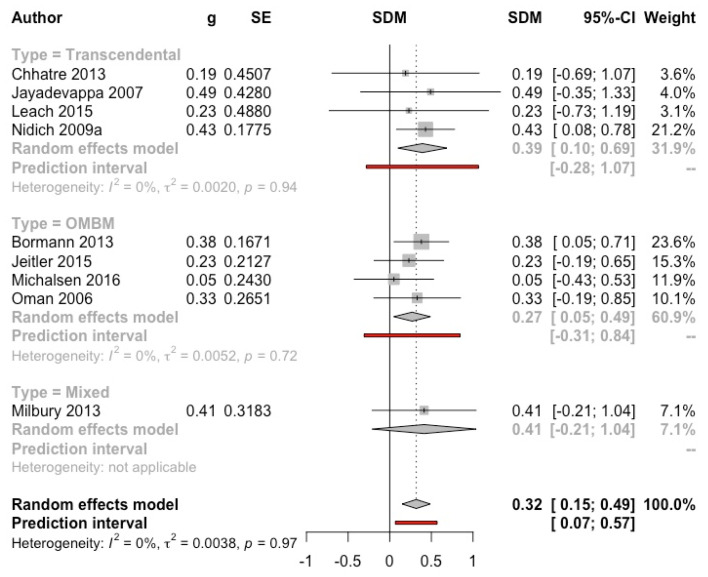
Effect on mental health-related quality of life (meditation vs. control). Predictive intervals are represented in red.

**Figure 11 ijerph-19-03380-f011:**
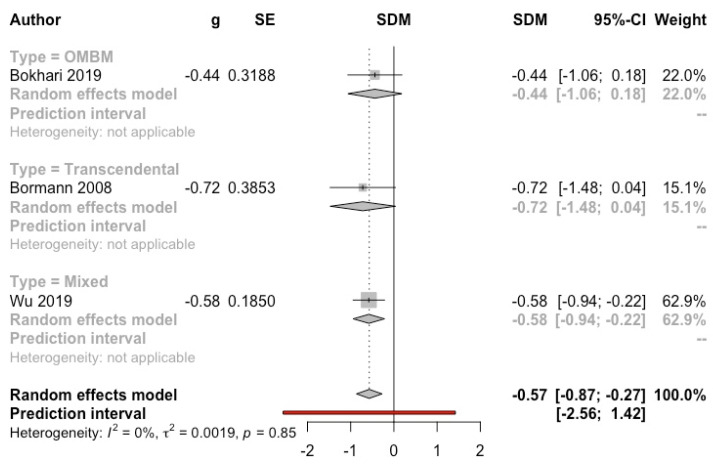
Effect on general psychopathology (meditation vs. control). Predictive intervals are represented in red.

## Data Availability

All data are included within the article and Appendix A.

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
