# Peer review of "Effectiveness of Mantra-Based Meditation on Mental Health: A Systematic Review and Meta-Analysis"

_ijerph, 2022, doi:10.3390/ijerph19063380_

Round 1

Reviewer 1 Report

Even with the care taken to organize the information in a supplementary document, I suggest that in Methods - item 2.1 the authors copy and paste the strategy from a database, to show the reader the syntax and what is the scope of the search (e.g., if by the presence of the terms in the titles/ abstracts or topics or as a word in the text).

I don't disagree that the "Cochrane risk-of-bias tool - RoB 2" is a great advance in the field, but it should be considered that its construct has greater application in assessing the quality of clinical trials. Issues such as blinding of participants is practically impossible in community-based studies, which involve practice. Thus, as the low quality of trials is one of the points highlighted in the review, I suggest weighting it by the characteristic of the instrument.

In relation to the meta-analysis, I point out a weakness that could be a little better detailed in the body of the text, and also indicated as a limitation of the research. Analysis of the "immediate post-intervention" and "follow up" try to answer different questions, the first being more focused on the observation of the immediate effects of the intervention, and the second on whether there was maintenance of the effects without the implementation of the intervention.

I recommend the presentation of a descriptive table (e.g., where the intervention was conducted, age group, gender, length of intervention, protocol, characteristics of the comparators) of the studies included in the body of the manuscript for a better understanding.

I notice that the conclusion is very generalized, in face of the mental health indicators evaluated. Thinking of a text more applied to decision making, I ask, with more specificity: on which indicators did the interventions produce more precise and large effects? In the summary, I suggest indicating the main effects, with confidence interval and percentage of I2 test, since greater heterogeneity in the indicators "stress" and "depression", respectively).

Reviewer 2 Report

This is an extremely well organized and implemented systematic review/meta-analysis following the current PRISMA guidelines. Excellent organization.

I am concerned with the default choice of a fixed effect model, and then only using random effects only on a significant Q test or an I-square greater than 50%. Current thinking on use of fixed effect or random effects models suggest deciding upfront (before analysis) whether a fixed effect or a random effects is most appropriate. Clearly, the conceptual heterogeneity of these studies along multiple parameters would suggest multiple population means therefore making the random effects model most appropriate.

Heterogeneity is to be anticipated in most cases, and whether fixed or random effects, the strategy is to try to explain the heterogeneity. This is done by stratification (which the authors have done, but mostly based on fixed effect results), or by meta-regression. Some of these investigations are included in the supplementary material.

I would suggest an a-priori specification (in the anticipation of heterogeneity under the belief a random effects model would be most appropriate due to conceptual heterogeneity), and specification of which factors are anticipated to be the sources of heterogeneity, and then use that a-priori specification as an analysis plan.

I definitely cannot recommend publishing on a fixed effect model, it would have to be random effects.

Last, once random effects results are estimated, then I strongly recommend (where possible, typically where studies > 3) is to report a prediction interval in addition to the confidence intervals. Prediction intervals are about dispersion, confidence intervals are about precision of studies in the meta-analysis.

I refer the authors to Borenstein et al, 2017 "Basics of meta-analysis: I-square is not an absolute measure of heterogeneity", that provides guidance relative to my points above. Also, this paper notes access to excel software program that will calculate the prediction intervals using the random effects results, or it may be obtained free (just send your email address in field in middle of webpage) at: Comprehensive Meta-Analysis Software (CMA)

Round 2

Reviewer 2 Report

Excellent! I've been a part of many reviews/meta-analyses; this is very well done.